# HP1γ Prevents Activation of the cGAS/STING Pathway by Preserving Nuclear Envelope and Genomic Integrity in Colon Adenocarcinoma Cells

**DOI:** 10.3390/ijms24087347

**Published:** 2023-04-16

**Authors:** Jorge Mata-Garrido, Laura Frizzi, Thien Nguyen, Xiangyan He, Yunhua Chang-Marchand, Yao Xiang, Caroline Reisacher, Iñigo Casafont, Laurence Arbibe

**Affiliations:** 1Université Paris Cité, INSERM, CNRS, Institut Necker Enfants Malades, 75015 Paris, France; 2The Nanomedicine Group, Institute Valdecilla-IDIVAL, 39011 Santander, Spain; 3Anatomy & Cell Biology Department, School of Medicine, University of Cantabria, 39011 Santander, Spain

**Keywords:** inflammatory bowel disease, epigenetics, HP1γ, inflammation, cGAS, STING, cytosolic DNA, nuclear envelop

## Abstract

Chronic inflammatory processes in the intestine result in serious conditions such as inflammatory bowel disease (IBD) and cancer. An increased detection of cytoplasmic DNA sensors has been reported in the IBD colon mucosa, suggesting their contribution in mucosal inflammation. Yet, the mechanisms altering DNA homeostasis and triggering the activation of DNA sensors remain poorly understood. In this study, we show that the epigenetic regulator HP1γ plays a role in preserving nuclear envelope and genomic integrity in enterocytic cells, thereby protecting against the presence of cytoplasmic DNA. Accordingly, HP1 loss of function led to the increased detection of cGAS/STING, a cytoplasmic DNA sensor that triggers inflammation. Thus, in addition to its role as a transcriptional silencer, HP1γ may also exert anti-inflammatory properties by preventing the activation of the endogenous cytoplasmic DNA response in the gut epithelium.

## 1. Introduction

Inflammatory bowel disease (IBD), including Crohn’s disease (CD) and ulcerative colitis (UC), is a chronic, relapsing inflammatory condition of the gastrointestinal tract [1]. It is a global health problem that predominantly affects young people between 15 and 30 years of age [2]. CD causes transmural inflammation and can involve any part of the gastrointestinal tract, although it most commonly affects the terminal ileum or the perianal region. In contrast, UC is typified by a mucosal inflammation limited to the colon and rectum [3]. The etiology of IBD involves a combination of genetic, epigenetic, environmental, microbial, and autoimmune factors, that often interact to cause the disease [4,5,6]. 

The innate immune response serves as the body’s first line of defense against invading pathogens, allowing for a rapid response to harmful stimuli. This response is carried out by various cell types, including epithelial cells, neutrophils, dendritic cells, monocytes, macrophages, and natural killer cells. The innate immune response is triggered by the recognition of microbial antigens, with the presence of DNA in the cell cytoplasm being one of the prominent antigens recognized [7]. The presence of DNA in the cytoplasm can indicate an ongoing viral or bacterial infection, as well as the result of endogenous double-stranded DNA, regardless of its sequence, including self-DNA [8]. This occurrence activates DNA sensors, such as the cGAS-cGAMP-STING pathway, which triggers inflammatory responses through the production of the second messenger cyclic GMP-AMP (cGAMP) [8]. Importantly, studies have shown increased expression of DNA sensors in the colon mucosa biopsy tissues of UC patients, with single-cell RNA sequencing data pointing to an increase in cGAS expression in both active and non-active UC colon epithelia [9,10]. These findings strongly suggest that a chronic or aberrant activation of DNA sensors contributes to the inflammation seen in IBD [11,12]. Yet, the mechanisms altering DNA homeostasis and triggering activation of DNA sensors in IBD remain unclear. 

Members of the HP1 family of proteins, which also includes HP1α and HP1β in mouse and human, are readers of the H3K9me2/3 histone modifications. They play key roles in the formation and maintenance of heterochromatin, thereby participating in transcriptional gene silencing [13]. In earlier studies, we have identified Heterochromatin Protein 1γ (HP1γ) as a regulator of inflammatory genes in response to enterobacteria, whose deficiency led to an IBD-like state, characterized by a colon dysbiosis and chronic inflammation [14,15].

The different members of the HP1 protein family play an important role in the maintenance of the nuclear envelope structure. These proteins bind to specific sites on the nuclear envelope [16,17,18,19,20], contributing to the generation of a polar protein-binding surface essential for the constitution of the nuclear envelope. In addition, they are important for the reassembly of the nuclear envelope after the process of mitosis [16,17,18,19,20]. Here, we show that HP1 maintains the integrity of the host’s nuclear envelope, which in turn prevents cytoplasmic DNA leakage and the subsequent activation of the pro-inflammatory cGAS/STING pathway in enterocytic cells.

## 2. Results

### 2.1. Loss of HP1γ Leads to Disruption in the Integrity of the Nuclear Envelope

To study the impact of HP1γ invalidation on nuclear structures, we carried out the deletion of the *Cbx3* gene in the colonic adenocarcinoma cell line TC7 using the CRISPR-Cas9 gene editing technique. Six clones were selected for this study, in which we found a potent inhibition in the detection of HP1γ by immunofluorescence microscopy (Appendix A). To gain a deeper understanding of this phenomenon, we conducted immunofluorescence microscopy and transmission electron microscopy (TEM) analyses (Figure 1). These experiments enabled us to observe significant changes in the nuclear envelope of TC7 *Cbx3* KO cells, including invaginations and spherical inclusions of varying sizes (Figure 1H, arrows). No such changes were observed in TC7 WT cells, which displayed a smooth and homogeneous nuclear envelope (Figure 1B). 

Using TEM, we observed that these invaginations in the nuclear envelope appeared as large double-membraned inclusions within the nucleus, often associated with significant amounts of heterochromatin in 65% of the cases observed (Appendix A, Figure 1H, arrows). They were in no case observed in the control cells, where the nuclear envelope was seen to surround the nucleus in a continuous manner (Figure 1G). Overall, these results showed that HP1γ deficiency induces a laminopathy. 

Based on the above observations, we concluded that TC7 *Cbx3* KO cells may have defects in cell viability. We found that despite exhibiting this phenotype from the time of collection (at passage 0), the cells began to slow their proliferation at passage 5 and entered senescence after 8 passages (Appendix A). Therefore, to ensure that our observations were not influenced by the cell senescence process, all experiments and results obtained in this study were conducted at passage 3 at the latest.

As shown in our previous work [15], the deletion of HP1γ leads to the production of progerin linked to an alteration in splicing fidelity (Figure 2). Progerin production was consistent with the laminopathy phenotype previously described (Figure 1). We compared progerin detection levels in TC7 WT cells (Figure 2A–C) and TC7 *Cbx3* KO cells (Figure 2D–F). In TC7 *Cbx3* KO cells, we detected progerin labeling in the nucleus (Figure 2D–F), whereas levels of this protein remained undetectable in WT cells (Figure 2A–C).

Additionally, we noted a multitude of small membranous changes within the nucleus of TC7 *Cbx3* KO cells (Figure 3). These changes took on various shapes, but always maintained their characteristic double-membrane envelope structure. They could be either irregular (Figure 3A) or spherical inclusions (Figure 3B–D) and were typically located in areas of gene repression, such as near the peripheral nuclear envelope (Figure 3A,B,D) or within large heterochromatin clusters (Figure 3C, asterisk). 

The structure and localization of these anomalies suggested that they were fragments of the nuclear envelope of the cell. This was supported by not only their double membrane conformation, but also their strong association with ribosomes (Figure 3B,D, arrows), as in the case of the nuclear envelope [21]. 

### 2.2. Cytoplasmic Chromatin Detection in TC7 Cbx3 KO Is Associated with DNA Damage and Progerin Production

Recurrent cytoplasmic DNA accumulations were detected in TC7 *Cbx3* KO cells, as shown by DAPI staining (Figure 4A). To further investigate these initial observations, we conducted a comprehensive electron microscopic analysis to search for chromatin inclusions in the cytoplasm of TC7 *Cbx3* KO cells. (Figure 4B–D). Indeed, our analysis revealed structures composed of a double membrane, similar in nature to the nuclear envelope, within chromatin inside (Figure 4B,C). These inclusions were frequently observed independently of the nuclear envelope and could be located at some distance from it. In addition, we observed electrodense structures in the cytoplasm that were similar in nature to free chromatin (Figure 4D, asterisk). These findings led us to believe that they could be fragments of nuclear chromatin that had become detached due to the genomic instability previously reported upon Drosophila HP1 loss [20]. We thus searched for the activation of the DNA damage response by performing immunolabeling of γH2AX followed by fluorescence microscopy analysis.

In TC7 *Cbx3* KO cells, we observed the induction of numerous DNA damage foci, which were labeled by γH2AX (Figure 5E). The analysis revealed that 93.8% of the cells were positive for these DNA damage foci (Figure 5G). In contrast, the DNA damage foci in TC7 wild-type cells were much smaller and were present in only 16.3% of the total cells (Figure 5B,G), possibly due to sporadic DNA damage [22,23]. Lastly, the phenotype was found reversible by the stable reintroduction of HP1γ expression in TC7 *Cbx3* KO cells. Indeed, the appearance of laminopathy as well as the formation of γH2AX-positive cells were significantly reduced (Appendix A). Additionally, the formation of γH2AX-positive cells remained at similar levels as in TC7 WT cells (Appendix A).

### 2.3. Cytoplasmic Chromatin in TC7 Cbx3 KO Is Associated with an Activation of the GAS/STING Pathway 

One of the primary mechanisms sensing the presence of cytosolic DNA is the cGAS/STING pathway [24]. In addition, progerin was previously shown to promote interferon-like responses via activation of the cGAS/STING pathway [25]. 

We therefore investigated the potential relationship between the loss of HP1γ and the activation of the cGAS/STING pathway. For this purpose, we carried out a study by immunofluorescence microscopy in which we compared the detection of cGAS and STING proteins in TC7 WT vs. TC7 *Cbx3* KO cells (Figure 6A,B). 

As demonstrated, TC7 WT cells exhibited a low expression of cGAS protein, which was mainly localized in the nuclei of positive cells (Figure 6A1–A3). In contrast, loss of HP1γ resulted in a detection of cGAS, which was mostly localized in the cytoplasm of the cells (Figure 6A4–A6), and 46% of the cells observed were quantified as positive, much higher than the 2% in TC7 WT cells (Figure 6B). Upon reintroduction of HP1γ in TC7 *Cbx3* KO cells, cytoplasmic STING levels returned to the basal levels seen in TC7 WT cells (Appendix A).

Based on these observations, we aimed to confirm the activation of this innate immune pathway by analyzing STING, the target of cGAS activation. As seen in the panel, STING was detected in only 1% of the TC7 wild-type cells (Figure 6C1–C3 and D), whereas in cells lacking HP1γ, a strong induction and clear labeling in the cytoplasm was observed, with 46% of cells being positive (Figure 6C4–C6 and D).

It is well-known that the cGAS/STING innate immunity pathway leads to the induction of pro-inflammatory factors, particularly those associated with interferon [26]. To further investigate this process, we conducted gene expression analysis of the interferon-induced antiviral proteins: interferon tetracyclopeptide (IFIT), IFIT1, IFIT3, and IFIT5, whose activity is associated with the cGAS/STING pathway [27,28,29]. Our findings indicated a significant transcriptional induction of these genes upon HP1γ depletion, as compared to WT cells. (Figure 6E–G). mRNA encoding for IL6, a cytokine playing an essential role in the survival of cancer cells with chromosomal instability, *via* the cGAS/STING pathway, was also robustly increased [30] (Figure 6H).

We next analyzed the expression levels of TGFβ, a protein downstream of STING and the type-I interferon (IFN) family. TGFβ is known to have an anti-inflammatory effect by inhibiting IFNs and STING in cancer models, which is crucial for tumor regression [31,32]. In our model, we observed a significant decrease in TGFβ expression levels in TC7 *Cbx3* KO cells compared to WT cells (Figure 6I). Altogether, these data show that HP1γ depletion in TC7 cell leads to an activation of the cGAS/STING inflammatory pathway. 

## 3. Discussion

Our results highlight the role played by HP1γ in maintaining nuclear membrane integrity and preventing uninduced DNA damage. We have found that depletion of HP1γ leads to genomic instability, characterized by the presence of DNA in the cytoplasm together with activation of the cGAS/STING pathway. These observations suggest an important role played by HP1 in the integrity of the nuclear envelope, with potential several mechanisms involved. Firstly, HP1 proteins were shown to localize at the nuclear periphery, where they serve as a scaffold for heterochromatin and thus may contribute to the maintenance of heterochromatin at this location [16], Moreover, dynamic binding sites for HP1γ have been detected at the nuclear envelope, reinforcing its consistency and stability [16]. Alternatively, depletion of HP1γ may promote the formation of double membrane substructures containing chromatin by affecting the expression of the lamins encoded by the *LMNA* gene. We previously showed that HP1γ ensures the proper splicing of *LMNA*, a gene which encodes for laminA and laminC, two important components of the nuclear envelope [15]. Likewise, depletion of HP1γ led to the activation of a weak and cryptic splice site at exon 11, resulting in the production of progerin at the expense of laminA [15]. Progerin is indeed a defective precursor of the laminA nuclear matrix protein, where the C-terminal cysteine, which is normally removed, is retained, and modified with a hydrophobic oligoisoprene chain [33,34]. Its incorporation into the lamina induces a laminopathy characterized by severe nuclear membrane deformations [35,36]. Finally, progerin itself has been shown to induce replication stress, leading to the accumulation of self-nucleic acids in the cytoplasm [37] and to cause unrepaired double-strand DNA damage. The latter may promote the loss of nuclear fragments or even free chromatin into the cytoplasm during the process of nuclear envelope reassembly after cell mitosis [38,39]. Based on these assumptions, we propose that HP1γ is a strong candidate for maintaining nuclear envelope integrity and genomic stability and that its loss may lead to cytoplasmic DNA leakage. Nevertheless, future experiments determining the nature of this cytoplasmic DNA will be necessary to confirm this hypothesis.

Endogenous cytoplasmic DNA can trigger cell-autonomous responses even in the absence of infection. Cytoplasmic DNA is a known contributor to sterile inflammation, which refers to inflammation in the absence of pathogenic infection. This type of inflammation is associated with many chronic age-associated conditions, including cancer, cardiovascular disease, and neurodegenerative disease [40,41]. Likewise, Li and colleagues demonstrated that certain stimuli, such as genotoxic stress, activate the cGAS/STING pathway, which can trigger unwanted cellular responses such as cellular senescence, inflammation, or cancer [40]. These phenomena are associated with the aging process. On the other hand, Kreienkamp et al. demonstrated that in Hutchinson-Gilford progeria syndrome (HGPS), one of the most well-documented premature aging syndromes, progerin-induced replication stress drives genomic instability by causing replication fork stalling and nuclease-mediated degradation. This event is accompanied by an upregulation of the cGAS/STING cytosolic DNA sensing pathway. [40,41]. Thus, aside from its function as an epigenetic silencer of inflammatory genes [42], HP1γ may also exert anti-inflammatory properties by preserving from a deleterious cytoplasmic DNA response. 

The uncontrolled activation of inflammatory pathways is a critical event in the onset and development of IBD and studies have reported increased activation of the cGAS/STING pathway in the colon epithelium [10]. However, the mechanism(s) of activation of the pathway remain unclear. As proposed by Ke et al., these mechanisms may involve cytoplasmic DNA sequences released from damaged mitochondria/nucleus inefficiently cleared by cellular ribonuclease activity or the transfer of microbiota products such as DNA or cyclic dinucleotides (where is the reference?Put the number). Building on our previous report that identified a large decrease in HP1γ expression in the epithelium of IBD patients [10], we propose that HP1γ may be part of the homeostatic mechanisms preserving the gut epithelium from the activation of DNA sensors by endogenous DNA species. More in-depth studies are required to elucidate the nucleic acid species activating the cGAS/STING pathway in the gut as well as their contribution to IBD exacerbation. 

## 4. Materials and Methods

### 4.1. Cell Cultures

TC7 cells (authenticated by ECACC) were used to generate the CRISPR/Cas9-mediated *Cbx3* cell line. The pSpCas9(BB)−2A-GFP (PX458) vector expressing Cas9 endonuclease (gift from Feng Zhang, Addgene plasmid #48138; Teddington, UK) was linked with a single-guide RNA (sgRNA) designed specifically for *Cbx3* gene. One Sequence guide (GAAGAAAATTTAGATTGTCC) was defined by ZiFiT Targeter Version 4.2 software (https://bio.tools/zifit). Insertion of the sequence guide was performed in the BbsI restriction site of the PX458 vector and checked by sequencing. Transfection in TC7 cells was performed by lipofectamine 2000 and single clones were selected by FACS according to the GFP signal. All experiments have been performed on cells at maximum pass 3 to avoid masking of the results due to the occurrence of senescence.

### 4.2. Immunofluorescence

TC7 cells were grown in coverslides at a maximum of 80% confluence. They were fixed 10 min RT with 3.7% PFA in PBS 1X and washed 3 times with PBS 1X. All samples were sequentially treated with 0.1 M glycine in PBS 1X for 15 min, and 0.5% Triton X-100 in PBS 1X for 15 min RT. Primary antibodies were then incubated with overnight at 4 °C, washed with 0.05% Tween-20 in PBS, incubated for 1 h in the specific secondary antibody conjugated with Alexa 488 or Cy3 (Jackson, Ely, UK), 15 min with DAPI (1μg/mL), washed in PBS, and mounted with the antifading medium VECTA- SHIELD^®^ (Vector laboratories). Anti-HP1γ (2MOD-IG6, Thermo Scientific, Illkirch-Graffenstaden, France), laminB1 (ab65986, Abcam, Cambridge, UK), Lamin A/C (4C11, Cell Signaling, Danvers, MA, USA), and histone H2AX phospho-Ser139 (Millipore-Upstate 05–636, MA, USA) were used as primary antibodies. Nuclei were stained using 4, 6-diamidino-2-phenylindole (DAPI, 62248, Thermo Scientific Illkirch-Graffenstaden, France).

Microscopy images were obtained with a ZEISS Apotome.2 (Zeiss, Leipzig, Germany), a structured illumination microscope, using a 63× oil (1.4 NA) objective. To avoid overlapping signals, images were obtained by sequential excitation at 488 and 543 nm in order to detect A488 and Cy3, respectively. Images were processed using ZEISS ZEN lite software (Zeiss, Leipzig, Germany).

### 4.3. Transmission Electron Microscopy

In TC7 cells, Transmission Electron Microscopy (TEM) was performed fixing overnight at 4 °C with 3.7% paraformaldehyde, 1% glutaraldehyde, in 0.1 M cacodylate buffer. After fixation, cells were pellet and contrasted with osmium tetroxide (OsO4) and uranyl acetate. The cells were then included in freeze-substitution medium for at least 3 days at −80 °C, dehydrated in increasing concentrations of methanol at −20 °C, embedded in Lowicryl K4M at −20 °C, and polymerized with ultraviolet irradiation. Ultrathin sections were mounted on nickel grids, stained with lead citrate and uranyl acetate and examined with a JEOL 1011 electron microscope.

### 4.4. Real Time Quantitative PCR

Total RNA was extracted using Trizol (TR-118, Molecular Research Center, Inc. Cincinnati, OH, USA) following the manufacturer’s instructions and DNAse treatment. RNA samples were quantified using a spectrophotometer (Nanodrop Technologies ND-1000). First-strand cDNA was synthesized by RT-PCR using a RevertAIT H Minus First Strand cDNA Synthesis kit (Thermo Scientific). qPCR was performed using the Mx3005P system (Stratagene, La Jolla, CA, USA) with automation attachment. The expression levels of the mRNA candidates (Origene, Rockville, MD, USA), IFIT1 (Fw: GCCTTGCTGAAGTGTGGAGGAA; Rv: ATCCAGGCGATAGGCAGAGATC), IFIT3 (Fw: CCTGGAATGCTTACGGCAAGCT; Rv: GAGCATCTGAGAGTCTGCCCAA), IFIT5 (Fw: CGTCCTTCGTTATGCAGCCAAG; Rv: CCCTGTAGCAAAGTCCCATCTG), IL6 (Fw: AGACAGCCACTCACCTCTTCAG; Rv: TTCTGCCAGTGCCTCTTTGCTG), TGF-β1 (Fw: TACCTGAACCCGTGTTGCTCTC; Rv: GTTGCTGAGGTATCGCCAGGAA) and GAPDH (Fw: GTCTCCTCTGACTTCAACAGCG; Rv: ACCACCCTGTTGCTGTAGCCAA) were determined by qPCR using gene-specific SYBRGreen (Takara, Shiga, Japan, RR820A)-based primers (0.4 µM). The specific sequence of the primers used should be requested. Expression was portrayed as mean ± SD.

### 4.5. Viability Assay by X-Gal Detection

The X-Gal detection protocol was carried out following that proposed by Debacq-Chainiaux et al., 2009 [43]. Briefly, the cells were fixed and incubated with the staining solution overnight at 37 °C. The next day, the cells were washed with PBS 1X 3 times 5 min at room temperature and their staining was analyzed using a brightfield microscopy.

### 4.6. HP1γ Recomplementation by Viral Transduction

The protocol for re-complementation was adapted from Harouz et al., 2014 [14]. Stable recombinants were obtained from TC7 Cbx3 KO cells by retroviral transduction according to the method of Nakatani and Ogryzko (Nakatani and Ogryzko, 2003), using the pOZ-N backbone retroviral vector. Human HP1γ cDNA was subcloned between the XhoI and NotI restriction sites of the pOZ-N vector. The stable presence of HP1γ was detected by immunofluorescence.

## Figures and Tables

**Figure 1 ijms-24-07347-f001:**
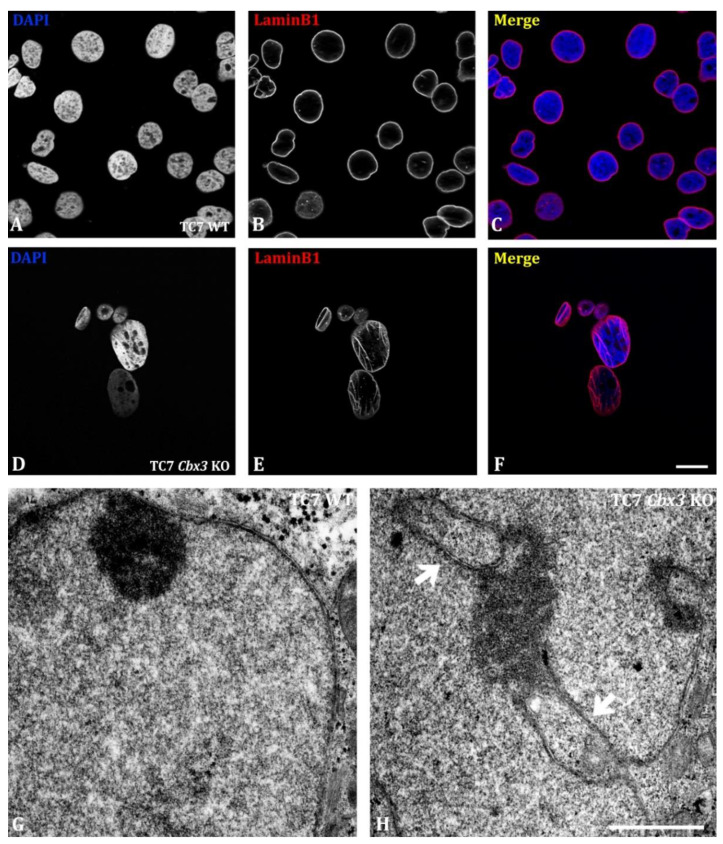
Alterations in the nuclear envelope in TC7 *Cbx3* KO cells. (**A**–**F**) Immunofluorescence of the nuclear envelope-associated protein LaminB1: large alterations can be seen in cells lacking HP1γ (**D**–**F**), whereas no abnormalities are seen in WT cells (**A**–**C**); (**G**,**H**) TEM analysis of nuclear envelope alterations in TC7 WT (**G**) or *Cbx3* KO (**H**) cells: double membrane inclusions can be observed in cells lacking HP1γ (**H,** arrows). *n* = 5 clones analyzed per condition, 2 plates per clone. Scale bar: (**A**–**F**) 20 µm; (**G**,**H**) 1 µm.

**Figure 2 ijms-24-07347-f002:**
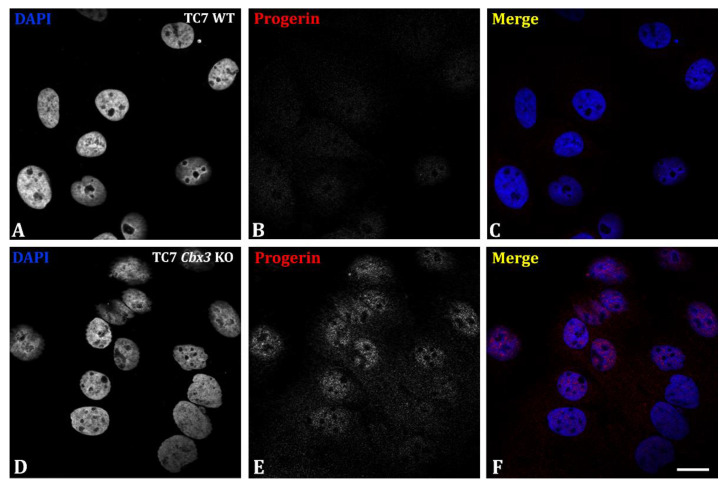
Analysis of progerin production upon depletion of HP1γ in TC7 cells. (**A**–**F**) Immunofluorescence study provides evidence for a detection of progerin in the nucleus of TC7 *Cbx3* KO cells (**D**–**F**) but not in TC7 WT (**A**–**C**). Images are representative of *n* = 5 clones (generated by 1 sgRNA) per condition, Scale bar: 25 µm.

**Figure 3 ijms-24-07347-f003:**
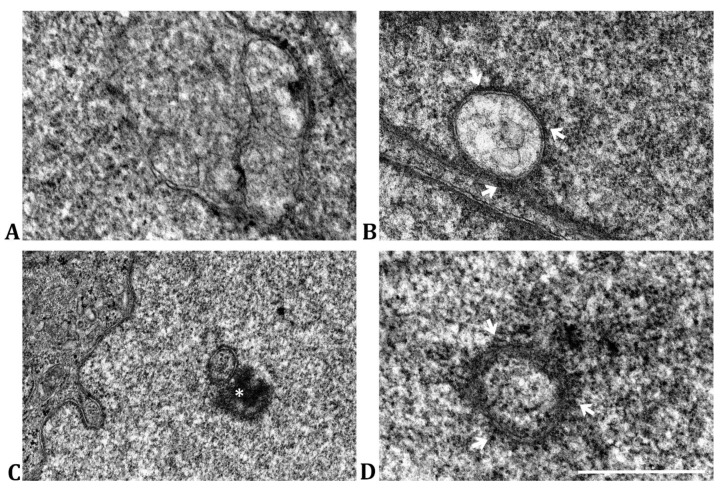
Ultrastructural analysis of membranous inclusions observed in TC7 *Cbx3* KO cells. (**A**) Irregular double membrane inclusion, observed next to the nuclear envelope. (**B**) Inclusion of the double spherical membrane, observed next to the nuclear envelope (arrows). The poorly electrodense nature of its contents can be appreciated. Arrows indicate the presence of ribosomes. (**C**) Inclusion of double spherical membrane, observed next to a large patch of heterochromatin (asterisk), suggesting the preference of these structures towards regions of gene repression. (**D**) Detail of small double spherical membrane inclusion, surrounded by ribosomes (arrows). *n* = 5 clones (generated by 1 sgRNA) per condition. Scale bar: 400 nm.

**Figure 4 ijms-24-07347-f004:**
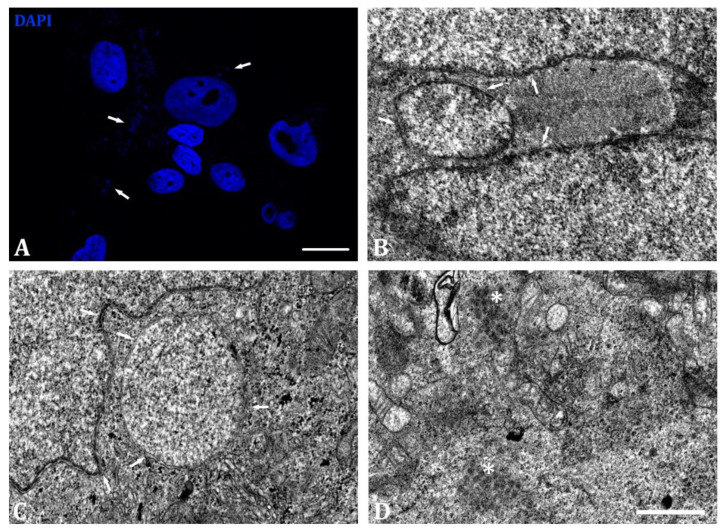
Ultrastructural analysis of membranous chromatin inclusions observed in the cytoplasm of TC7 *Cbx3* KO cells. (**A**) Evidence for the presence of DNA-containing structures in the cytoplasm by staining with DAPI. (**B,C**) Examples of chromatin-containing double-membraned cytoplasmic inclusions (double membrane featured by arrows) observed in TC7 *Cbx3* KO cells. While these inclusions are located close to the nuclear envelope, they are not in direct contact with it. (**D**) Detail of free chromatin structures (asterisk) observed in the cytoplasm of cells lacking HP1γ. These structures are found free in the cytoplasm, with no preference for any organelle or nuclear substructure. *n* = 5 clones (generated by 1 sgRNA) per condition. Scale bar: (**A**) 25 µm; (**B**–**D**) 400 nm.

**Figure 5 ijms-24-07347-f005:**
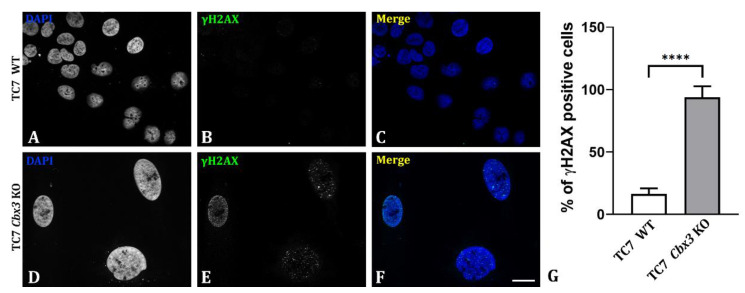
Analysis of the induction of DNA damage foci upon loss of HP1γ in TC7 cells. (**A**–**F**) Immunofluorescence study for detecting DNA damage foci, defined by γH2AX detection in TC7 WT (**A**–**C**) and TC7 *Cbx3* KO cells (**D**–**F**). (**G**) Quantitative analysis of γH2AX upregulation after loss of HP1γ in TC7 enterocytic cells. *n* = 5 clones (generated by 1 sgRNA) per condition, 2 plates per clone (in WT, different plates were analyzed together). Values are mean ± SD. **** *p* < 0.0001 (*t*-test and non-parametric, two-tailed *p* value). Scale bar: 25 µm.

**Figure 6 ijms-24-07347-f006:**
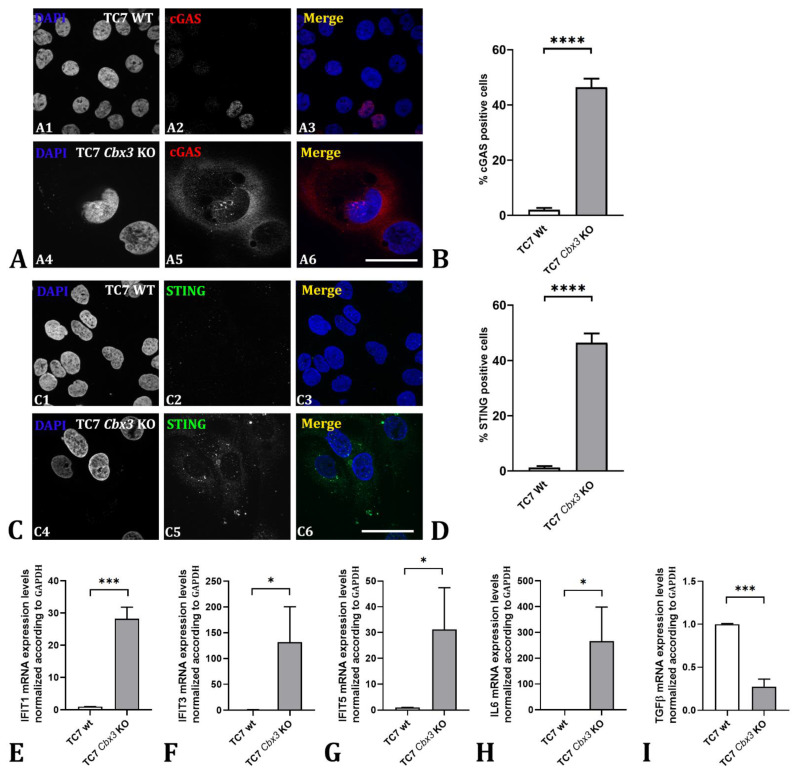
Analysis of the c-GAS-inflammatory response upon depletion of HP1γ in TC7 cells: (**A**) Detection of cGAS by immunofluorescence microscopy: the cGAS signal was weakly visible in TC7 WT cells (**A1**–**A3**), and localized in the cell nucleus, whereas expression is triggered in TC7 *Cbx3* KO cells and principally detected in the cytoplasm (**A4**–**A6**). (**B**) Quantitative analysis of the number of cGAS positive cells. (**C**) STING detection by immunofluorescence microscopy: STING protein remains at undetectable levels, based on our parameters in TC7 WT cells (**C1**–**C3**), while in TC7 *Cbx3* KO cells (**C4**–**C6**), its presence in the cytoplasm was markedly induced. (**D**) Quantitative analysis of the number of STING positive cells. (**E**–**I**) Q-PCR analysis of the corresponding genes associated with the inflammatory response. *n* = 3 clones per condition (generated by 1 sgRNA), 3 plates per clone (different plates were analyzed together). Values are mean ± SD. * *p* < 0.05; *** *p* < 0.001; **** *p* < 0.0001 (*t*-test and non-parametric, two-tailed *p* value). Scale bar: 50 µm.

## Data Availability

The data presented in this study are available in the article.

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
