# Peer review of "HP1γ Prevents Activation of the cGAS/STING Pathway by Preserving Nuclear Envelope and Genomic Integrity in Colon Adenocarcinoma Cells"

_ijms, 2023, doi:10.3390/ijms24087347_

Round 1

Reviewer 1 Report

HP1γ prevents activation of the cGAS/STING pathway by pre-2 serving nuclear envelope and
genomic integrity in colon adeno-3 carcinoma cells.
Mata-Garrido J. et al. identified a novel mechanism that mantains nuclear envelop stability and thus
genome integrity through the expression of HP1γ in colonic cells. Knocking out HP1γ leads to aberrant
nuclear envelop leading to DNA damage and nuclear DNA leaking to the cytoplasm which in turns
activates cGAS/STING pro-inflammatory pathway. The authors show this by using
immunofluorescence and electron microscopy experiments.
The findings are novel and significant in that they identify a new mechanism that regulates
inflammation. However, additional experiments are needed to increase the quality of the study and for
publication:
Major comments:
-When authors in the legend refer to clones, do they refer to clones from one sgRNA? or do they refer
to 5 different sgRNA clones? they should clarify this. If only one sgRNA has been used, an additional
sgRNA should be used showing similar results at least for some of the experiments.
-and are typically located in areas of gene repression, such as near the periph-115 eral nuclear
envelope (Figure 3A,B,D) or within large heterochromatin clusters 116 (Figure 3C, asterisk).
If possible the authors should test this with transcriptional repression markers by immunofluorescence.
Additionally, the authors should quantify how many invaginations are next to heterochromatin and how
many are not and show more examples at least in supplementary material.
- Figure 4A. It would be good to confirm it is nuclear DNA. If as in the abstract is mentioned, this DNA
is damaged, perhaps by using gH2AX staining.
-These cells displayed an ab-77 normal nucleus morphology, visible by transmitted light microscopy
(data not 78 shown).”
The authors should show this in supplementary or remove the statement.
-Fig. 6: it looks that the cell size in A1-3 and A4-A6 is not the same. It would be good to show more
cell number and quantify the number of cells with cytoplasmic cGAS and STING staining.
-The authors suggest that inflammation is activated upon HP1γ loss. It would be good to measure
inflammation markers by qPCR, WB or IF.
Minor comments:
-“...spherical inclusions of varying sizes (Figure 1E, arrows).” Fig 1E does not have arrows. Do the
authors refer to 1H?
- It would help if the author used a different color to call the panels in Fig. 3.
- Additionnally, we noted a multitude of small membranous changes within 111 the nucleus of TC7
Cbx3 KO cells (Figure 3). These changes took on various 112 forms ????, but always maintained their
characteristic double-membrane enve-113 lope structure. They can be either irregular (Figure 3A) or
spherical (Figure 3B-114 D) and are typically located in areas of gene repression, such as near the
periph-115 eral nuclear envelope (Figure 3A,B,D) or within large heterochromatin clusters 116 (Figure
3C, asterisk). This is unclear. In the legend you talk about inclusions.” It seems that the authors sent
a version that was still being edited.

Author Response

Reviwer1

Major comments:

  1. When authors in the legend refer to clones, do they refer to clones from one sgRNA? or do they refer to 5 different sgRNA clones? they should clarify this. If only one sgRNA has been used, an additional sgRNA should be used showing similar results at least for some of the experiments.

While we agree that using different sgRNAs would further document that our phenotype is not due to a particular sgRNA, such an experiment would unfortunately not be feasible within the time frame allowed for the revision of the manuscript. However, we have now performed rescue experiments by re-introducing HP1γ expression into the TC7 Cbx3 KO model, which demonstrate a nearly complete reversal of the phenotype. Therefore, our initial observations are entirely linked to HP1γ depletion (refer to new Supplementary Figure 4).

  1. “and are typically located in areas of gene repression, such as near the periph-115 eral nuclear envelope (Figure 3A,B,D) or within large heterochromatin clusters 116 (Figure 3C, asterisk).”If possible the authors should test this with transcriptional repression markers by immunofluorescence. Additionally, the authors should quantify how many invaginations are next to heterochromatin and how many are not and show more examples at least in supplementary material.

We agree with the referee’s comment.  This has been included in Sup. Fig. 2, along with more examples by TEM and immunofluorescence. Unfortunately, we could not perform the combination of LaminB1 and heterochromatin markers because the available antibodies are made in the same species.

  1. Figure 4A. It would be good to confirm it is nuclear DNA. If as in the abstract is mentioned, this DNA is damaged, perhaps by using gH2AX staining.

We thank the Reviewer for this suggestion. DNA visualized in the cytoplasm more likely origins from the nucleus since we observed that some (but not all) of the DAPI positive foci in the cytoplasm were also positive for H2AX, as shown in the image (provided for reviewer only).  But we believe that this important question calls for a long-term project that we aim to  initiate by implementing correlative light-electron microscopy (CLEM) in order to characterize the histones present in DAPI-positive foci in the cytoplasm. We hope this image will be sufficient to show to the Reviewer the possible origin of these DAPI-positive foci. We hope the reviewer will find the answer satisfactory

  1. “These cells displayed an ab-77 normal nucleus morphology, visible by transmitted light microscopy (data not 78 shown).” The authors should show this in supplementary or remove the statement.

We thank the Reviewer  for pointing this out. While we were referring to the researcher's observation under the microscope, we understand the confusion and have eliminated the phrase. We invite the reviewer to take a look at Supplementary Figure 2, where we performed X-Gal staining and the morphology we referred to can be seen.

  1. 6: it looks that the cell size in A1-3 and A4-A6 is not the same. It would be good to show more cell number and quantify the number of cells with cytoplasmic cGAS and STING staining.

This has been adressed  in Figure 6 of the revised manuscript

  1. The authors suggest that inflammation is activated upon HP1γ loss. It would be good to measure inflammation markers by qPCR, WB or IF.

We thank the Reviewer for this suggestion. The  qPCRs data are now included in Figure 6 of the revised manuscript

Minor comments:

  • “…spherical inclusions of varying sizes (Figure 1E, arrows).” Fig 1E does not have arrows. Do the authors refer to 1H?

The Reviewer is right. We apologize for this mistake that has now been corrected.

  • It would help if the author used a different color to call the panels in Fig. 3.

We have now renamed the panels by mentioning the lettering outside the photo, which has improved its visibility.

  • “Additionnally, we noted a multitude of small membranous changes within 111 the nucleus of TC7 Cbx3 KO cells (Figure 3). These changes took on various 112 forms ????, but always maintained their characteristic double-membrane enve-113 lope structure. They can be either irregular (Figure 3A) or spherical (Figure 3B-114 D) and are typically located in areas of gene repression, such as near the periph-115 eral nuclear envelope (Figure 3A,B,D) or within large heterochromatin clusters 116 (Figure 3C, asterisk). This is unclear. In the legend you talk about inclusions.”

The Reviewer is right. We apologize for these mistakes in the legends have now been corrected.

  • It seems that the authors sent a version that was still being edited.- “cytoplasm of the cells (Figure 6 A3-A6).” Do the authors mean A4?

The Reviewer is right. The mistake has been corrected.

  • In the methods section there is a “Real time quantitative PCR.” Section. However, there are no results showing and qPCR.

The Reviewer is right. The mistake has been corrected. The qPCRs have been included in Figure 6.

Reviewer 2 Report

The authors of this study report the implication of HP1γ in the prevention of inflammatory bowel disease (IBD) by using TC7 cells and HP1gamma KO (Knock Out) cells as a model. They show that HP1γ loss leads to nuclear membrane deformation, DNA damage and expression of cGAS/STING pathway components. Although an interesting concept, the implication of HP1γ in the development of inflammatory diseases, there are issues with controls, KO cell characterization, substantiation of HP1γ implication and potentially missing parts from the text that must be experimentally mitigated.

1) Introduction 62 to 70: The authors quote and reference citations 16-17 for HP1γ function in nuclear dynamics 

(16. Li T, Chen ZJ. The cGAS-cGAMP-STING pathway connects DNA damage to inflammation, senescence, and cancer. J Exp Med. 359 2018 May 7;215(5):1287-1299. doi: 10.1084/jem.20180139. 360

17. Motwani M, Pesiridis S, Fitzgerald KA. DNA sensing by the cGAS-STING pathway in health and disease. Nat Rev Genet. 2019 361 Nov;20(11):657-674. doi: 10.1038/s41576-019-0151-1.).

However, these papers concern the cGAS-STING system and do not mention HP1 family members even once. More specific to HP1γ and its function citations are required.

2) Figure 1: Quantification and biological replicates of this experiment must be reported. Although the image is clear, it still shows 5-6 cells (in the KO mutants). 

Furthermore, as these KO clones result in serious nuclear envelope defects, DNA damage e.t.c. (and they are different as mentioned in line 102-103) their viability in culture must be shown with any kind of assay compared to WT.

3) Lines 111-113Do the question marks (??????) signify something? Is there something missing? 

4) Figure 4A: As CCFs (cytoplasmic chromatin fragments) are characterized by heterochromatin and DAPI is a general chemical reagent that can potentially give rise to background noise, the presence of CCFs in KO cells must be also documented with a heterochromatin marker (e.g. H3K27me3; Ivanov A et al, JCB 2013) in order to understand the extend of this phenomenon.

5) The authors reports in the material and methods section that they conducted RT-PCR measurements of IFIT1, IFIT3, IFIT5, IL6 and TGF-β1 (Section 4.4). However, the respective figures and text are missing in my version of their text. These measurements are essential as they ‘ll show the induction of the system.

6) My most critical remark is that the authors must substantiate HPγ implication by conducting phenotype rescue experiments in their KO cell lines. So, the authors must show that exogenous HP1γ expression in KO cells rescues nuclear envelope integrity (immunofluorescence after HP1γ expression in KOs) and cGAS signaling (e.g. RT-PCR after HP1γ expression in KOs)

7Materials and methods section and throughout the text: a) Gene names as Cbx3 should be mentioned in inclined lettering. b) As CCF (cytoplasmic chromatin fragments) and activation of cGAS-STING system is also an aspect of cellular senescence the authors should state the passage that their cells were acquired and used. c) oligos for RT-PCR (sequence or accession number) should be reported.

8The text needs proofreading to avoid syntax and typographical errors (e.g. line 268 grwon)

Author Response

Reviwer2

1) Introduction 62 to 70: The authors quote and reference citations 16-17 for HP1γ function in nuclear dynamics 

(16. Li T, Chen ZJ. The cGAS-cGAMP-STING pathway connects DNA damage to inflammation, senescence, and cancer. J Exp Med. 359 2018 May 7;215(5):1287-1299. doi: 10.1084/jem.20180139. 360

  1. Motwani M, Pesiridis S, Fitzgerald KA. DNA sensing by the cGAS-STING pathway in health and disease. Nat Rev Genet. 2019 361 Nov;20(11):657-674. doi: 10.1038/s41576-019-0151-1.).

However, these papers concern the cGAS-STING system and do not mention HP1 family members even once. More specific to HP1γ and its function citations are required.

The Reviewer is right. We apologize for this mistake.  New references have been added.

2) Figure 1: Quantification and biological replicates of this experiment must be reported.

The informations requested on the biological replicates have now been included (Figure XX of the revised masnucript)

 Although the image is clear, it still shows 5-6 cells (in the KO mutants). 

Furthermore, as these KO clones result in serious nuclear envelope defects, DNA damage e.t.c. (and they are different as mentioned in line 102-103) their viability in culture must be shown with any kind of assay compared to WT.

We agree with the referee comment. We have included a viability test that we performed, the incorporation of X-Gal, as a senescence marker (Sup.Fig.3). In addition, we have specified when its appearance takes place in the text and we note that we use cells always before the senescence appearance, which is pass 5. All experiments were carried out at pass 3 maximum.

We agree with the referee's comment. We have included a viability test that we performed, which involved the incorporation of X-Gal as a senescence marker (New Supplementary Figure 3). Additionally, we have specified in the text when the appearance of senescence starts to take place, which is at passage 5. It is worth noting that all experiments were carried out with cells at a maximum of passage 3

3) Lines 111-113: Do the question marks (??????) signify something? Is there something missing? 

We apologize for this error. We suspect that the issues were caused by the use of different computer formats (Mac and PC), and we cannot find any other explanation. We have taken steps to correct the problem

4) Figure 4A: As CCFs (cytoplasmic chromatin fragments) are characterized by heterochromatin and DAPI is a general chemical reagent that can potentially give rise to background noise, the presence of CCFs in KO cells must be also documented with a heterochromatin marker (e.g. H3K27me3; Ivanov A et al, JCB 2013) in order to understand the extend of this phenomenon.

We thank the Reviewer for this suggestion. This article shows that CCFs are positive for both γ-H2AX-positive and H3K27me3.  We indeed observed that some (but not all) of the DAPI positive foci in the cytoplasm were also positive for H2AX, as shown in the image (provided for reviewer only). But we believe that this important question calls for a long-term project that we aim to initiate by implementing correlative light-electron microscopy (CLEM) to correlate EM features to the histone marks.

We hope this image will be sufficient to show to the Reviewer the possible origin of these DAPI-positive foci. We hope the reviewer will find the answer satisfactory

5) The authors reports in the material and methods section that they conducted RT-PCR measurements of IFIT1, IFIT3, IFIT5, IL6 and TGF-β1 (Section 4.4). However, the respective figures and text are missing in my version of their text. These measurements are essential as they ‘ll show the induction of the system.

We thank the Reviewer for this suggestion. The  qPCRs data are now included in Figure 6 of the revised manuscript

6) My most critical remark is that the authors must substantiate HPγ implication by conducting phenotype rescue experiments in their KO cell lines. So, the authors must show that exogenous HP1γ expression in KO cells rescues nuclear envelope integrity (immunofluorescence after HP1γ expression in KOs) and cGAS signaling.

We fully agree with the Reviewer’s comment. we have now performed rescue experiments by re-introducing HP1γ expression into the TC7 Cbx3 KO model, which demonstrate a nearly complete reversal of the phenotype (H2AX, cGas signaling). Therefore, our initial observations are entirely linked to HP1γ depletion (refer to new Supplementary Figure 4).

7Materials and methods section and throughout the text: a) Gene names as Cbx3 should be mentioned in inclined lettering. b) As CCF (cytoplasmic chromatin fragments) and activation of cGAS-STING system is also an aspect of cellular senescence the authors should state the passage that their cells were acquired and used. c) oligos for RT-PCR (sequence or accession number) should be reported.

Thank you very much for your comment. It has been taken into account, and the text has been corrected accordingly

8The text needs proofreading to avoid syntax and typographical errors (e.g. line 268 grwon)

The text has now been reviewed by external readers, and we believe that the syntax and typographical errors have been corrected

Round 2

Author Response

HP1γ prevents activation of the cGAS/STING pathway by pre-2 serving nuclear envelope and  genomic integrity in colon adeno-3 carcinoma cells. While the authors have addressed most of the reviewer comments, there are further revisions that  should be addressed.

1.- The MS has been improved by showing that reconstitution of HP1γ in the KO cells reverse the KO phenotype in Supplementary Fig. 6. However, the authors only showed reconstituted pictures. Images from parallel experiments with EV reconstitution should be included and the quantification should be compared to EV from that experiment.

We agree with the reviewer's comment and, in fact, we carried out the infection experiment with the empty vector in parallel when we designed all conditions.

If we didn't include the quantitative analysis of the TC7 Cbx3 KO + empty vector cells initially, it is because these cells died during the process of recomplementation with the empty vector after undergoing the CRISPR process. The reviewer should note that cells lacking HP1g began to slow their proliferation at passage 5 and entered senescence after 8 passages, which makes the time frame for obtaining a recomplemented cell line too short. Recomplementation with HP1 gamma prevented the development of senescence, allowing us to describe the rescue phenotype, as shown in Supplementary Figure 4.

Additionally, since they have only used one sgRNA clone, they should indicate in all the  legends as well as text that they are different cell clones from one sgRNA as it can be misunderstood  by the reader.

According to the reviewer's suggestion, we have changed all the legends, including that the clones were obtained from 1 single sgRNA.

Moreover, in the methods, the authors report the use of two sgRNA sequences. However, they  only show data from one sgRNA. If they do have a second sgRNA, the authors should include data  from the second sgRNA.

The reviewer is correct and we apologize for the error. We had included both sgRNAs because we performed the CRISPR experiment with both, but only succeeded in obtaining TC7 Cbx3 KO cells with one of them. Therefore, we have removed the error from the materials and methods.

2.- Supplementary Fig.2 : It is understandable that co-IF cannot be performed due to antibody species.  However, in Supplementary Fig.2 the authors should indicate in the legend which marker they used for heterochromatin, even if it is DAPI staining.  Additionally, a WT cell immunofluorescence should be included as a negative control. 

The corrections suggested by the reviewer have been taken into account and implemented

3.- Regarding comment 3, if the authors do not want to show the image with additional negative  controls in the main MS, they should re-write the abstract, discussion and text to better reflect what  the current results are showing. The authors do not confirm that the cytoplasmic DNA is damaged or  nuclear DNA in the current version.

The reviewer's comments have been taken into account and the following corrections have been made as followed:

1) In the abstract, any mention that the cytoplasmic DNA comes from the nucleus has been removed.

2) Between lines 147 - 149: "These findings led us to believe that they could be fragments of nuclear chromatin that had become detached due to the genomic instability previously reported upon Drosophila HP1 loss [22]".

We believe that there is no need to change that sentence, as we are stating a hypothesis based on our considerations. At no point do we say that the cytoplasmic DNA comes from the nucleus. We suggest that they could be nuclear fragments, based on the paper cited.

3) Lines 235-237: "presence of damaged DNA in the cytoplasm"; we have eliminated "damaged", since we do not demonstrate that it is damaged, but we are sure that it is DNA.

4) Lines 243-244: we have replaced "endogenous DNA leakage", by "formation of double membrane substructures containing chromatin". This phrase can be used since we have demonstrated by TEM the presence of these structures (Figure 4).

5) Discussion 256-263: We propose that based on the literature, the presence of progerin may result in the accumulation of nucleic acids and chromatin in the cytoplasm, as well as unrepaired DNA. Since this is a hypothesis that we are considering and proposing in the discussion, We believe that there is no need to change that sentence, However, we have added the sentence: "Nevertheless, future experiments determining the nature of this cytoplasmic DNA will be necessary to confirm this hypothesis". This addition clarifies that we recognize the need for further research to confirm our hypothesis.

In all other instances where 'cytoplasmic DNA' is mentioned in the paper, its origin is not specified.

Reviewer 2 Report

The authors experimentally responded to my concerns.

They only have to fill out the information on all supplements in the space provided (lines 364-365)

Author Response

Thank you very much for your comment. The information concerning the supplementary figures has been answered, as the reviewer has indicated. 

Round 3

Reviewer 1 Report

The authors have reviewed all of this reviewer's comments and improve the quality and accuracy of the text and results. 

One minor comment in the methods section:

"Transfection in TC7 cells was performed by lipofectamine 2000 and singles clones for each sequence guide were selected by FACS according to the GFP signal. "

Since the authors only describe one sgRNA, the sentence should be adjusted. 

Author Response

Thank you very much for your comment. The sentence has been corrected as the reviewer indicates: "Transfection in TC7 cells was performed by lipofectamine 2000 and singles clones were selected..."
